Subject Areas:
behaviour/cognition/psychology

Keywords:
face processing, individual differences, subprocesses, personnel selection

Author for correspondence:
Meike Ramon
e-mail: meike.ramon@gmail.com

# Differences between and within individuals, and subprocesses of face cognition: implications for theory, research and personnel selection

Matthew C. Fysh[1], Lisa Stacchi[2] and Meike Ramon[3]

[1]School of Psychology, University of Kent, Canterbury, Kent, UK
[2]iBM Lab, Department of Psychology, and [3]Applied Face Cognition Lab, Department of Psychology, University of Fribourg, Faucigny 2, 1700 Fribourg, Switzerland

 MCF, 0000-0002-3812-3749; MR, 0000-0001-5753-5493

Recent investigations of individual differences have demonstrated striking variability in performance both *within* the same subprocess in face cognition (e.g. face perception), but also *between* two different subprocesses (i.e. face *perception* versus face *recognition*) that are assessed using different tasks (face *matching* versus face *memory*). Such differences between and within individuals between and within laboratory tests raise practical challenges. This applies in particular to the development of screening tests for the selection of personnel in real-world settings where faces are routinely processed, such as at passport control. The aim of this study, therefore, was to examine the performance profiles of individuals within and across two different subprocesses of face cognition: face perception and face recognition. To this end, 146 individuals completed four different tests of face matching—one novel tool for assessing proficiency in face perception, as well as three established measures—and two benchmark tests of face memory probing face recognition. In addition to correlational analyses, we further scrutinized individual performance profiles of the highest and lowest performing observers identified *per test*, as well as *across all tests*. Overall, a number of correlations emerged between tests. However, there was limited evidence at the individual level to suggest that high proficiency in one test generalized to other tests measuring the same subprocess, as well as those that measured a different subprocess. Beyond emphasizing the need to honour inter-individual differences

through careful *multivariate* assessment in the laboratory, our findings have real-world implications: combinations of tests that most accurately map the task(s) and processes of interest are required for personnel selection.

## 1. Introduction

Studies of human face processing are of practical relevance to identity verification in a number of real-world settings (see [1]). In airports, for example, passport officers perform unfamiliar face *matching*, whereby they compare the face of each traveller to a passport photograph to verify their identity (e.g. [2]). Similarly, in forensic settings, police officers may be required to match a high-quality photograph of a suspect to lower quality surveillance footage to establish their involvement in a crime (e.g. [3–5]). Conversely, eyewitnesses may also be called upon by police to *recognize* the face of a suspect that they have seen previously, but with whom they are otherwise unfamiliar (e.g. [6]).

Experimental investigations into the distinct subprocesses of face cognition, i.e. perception and recognition, have revealed that matching and recognition of unfamiliar faces are generally challenging tasks, which can be performed relatively more proficiently with personally familiar faces (for reviews, see [7–9], respectively). Even in tasks designed to maximize accuracy, observers can exhibit error rates of 10–20% in face matching (e.g. [10–12]), and around 30% errors in tests of face memory (e.g. [6,13]). Such error rates raise concerns surrounding the reliability of facial identity verification in operational settings. These concerns are further compounded by the finding that groups of professionals who are responsible for facilitating identity verification often perform comparably (poorly) to student novices (see [2,14–17]).

Attempts to train performance in these tasks of identity verification have been met with limited success (e.g. [18–21]). An alternative approach to minimizing errors in applied settings, therefore, might be to recruit individuals who already possess an innate proficiency for processing unfamiliar faces (see [22,23]). Studies show, for instance, that some individuals consistently achieve near-ceiling accuracy rates in the same matching tasks performed over consecutive days [24], and are relatively unaffected by increases in task difficulty [13,25,26]. Such individual differences have also been observed within professional populations, where accuracy ranges from chance to perfect [2,27]. These findings converge with a growing research interest in the abilities of so-called super-recognizers [28], who display high proficiency in tests of face matching and/or face memory, suggesting superior face perception and/or recognition abilities (see [23,29,30]).

Based on this research, a sensible strategy might be to recruit such individuals with enhanced face processing ability for real-world occupations in which face matching and recognition is crucial. Yet, it is unclear how such individuals should be recruited. Given that people typically possess limited insight into their own ability to process faces [31–33], self-selection is unlikely to provide a reliable basis for the recruitment of personnel. An alternative solution might, therefore, be to develop batteries of tests that can identify individuals who are highly proficient at face processing (e.g. [22,26,31,34]).

In principle, this proposition makes good sense. Yet in practice, it is difficult to establish which tests should be included in such a battery. One challenge for screening tests, for instance, is to employ a range of measures that are sufficiently challenging so as to amplify differences between high performers, and thus provide a means of identifying observers who consistently achieve high accuracy across subprocesses and their respective tasks. Indeed, the difficulty of selecting tests that are appropriate for this purpose has been demonstrated in several studies, in which the employed tests suffered from either ceiling effects (e.g. [26,34]) or floor effects [31], and thus lack the sensitivity to identify high-performing individuals.

A second obstacle that must be overcome when constructing batteries of tests concerns both the degree of overlap between laboratory measures of face processing ability, as well as the extent to which these assessments overlap with the demands of processing faces in the real world [23,29,35]. This pertains both to the subprocess of interest (e.g. perception versus recognition, assessed via tasks of face matching versus face memory), as well as procedural aspects (e.g. the number and frequency of trials, stimulus presentation duration and the time interval between acquisition of two to-be-matched face photographs [23,36–38]. Since different tests of face cognition designed in laboratory settings typically vary with respect to subprocesses measured and procedural details, it is unsurprising that a large degree of intra-subject variability is observed even within high-performing

individuals ([31]; see also [25]). Consequently, multiple tasks should ideally be used to measure a single subprocess of interest, which should not be conflated with a related, but different subprocess.

From a theoretical perspective, addressing these practical challenges holds some informative value. For instance, research has suggested the existence of a face-specific cognitive component which underpins performance across multiple subcomponents of face processing [33,39,40], with the exception of human face detection [11,41]. While there is evidence that face matching and face memory performance may depend on overlapping cognitive mechanisms [11,37], other work suggests that the underlying processes can also operate independently in average and high performers ([31,41]; see also [25]). Thus, which aspects determine performance across tests of the same and different subprocesses of face cognition remains an open question. Two scenarios seem entirely plausible: (i) individuals who achieve high performance across multiple tests of face matching could also excel at different tests of face memory (i.e. enhanced face perception *and* recognition); (ii) high performance across tests of face matching could coincide with average performance across tests of face memory (i.e. enhanced face perception *only*). The latter pattern has been hinted at in studies that find correlations between two or three different tests of face matching [12,25,34,37]. However, it remains difficult to fully determine whether enhanced performance remains stable across a battery of face matching tasks that differ regarding, for example, the number of to-be-matched identities, or variability across stimuli presented.

In line with this reasoning, Stacchi *et al.* [42] recently provided normative data for two difficult tasks of face matching: the Yearbook Task (YBT; [43]) and the Facial Identity Card Sorting Task (FICST; [44]). Both of these tasks require observers to match images of unfamiliar faces across substantial variability in appearance due to ageing, facial paraphernalia and ambient conditions. Importantly, performance in the YBT and FICST was highly correlated. Moreover, performance in the YBT correlated reliably with (i) inverse efficiency computed for two tests of pairwise face matching (see [17]) and (ii) performance in the long version of the Cambridge Face Memory Test (CFMT+; [28]), which has been used as a benchmark measure of face recognition for detecting super-recognizers [23].

These findings imply that the YBT and the FICST together comprise useful predictors of face perception, as well as general face recognition ability as measured by the CFMT+. However, the extent to which the YBT and the FICST may predict performance in tasks of face matching approximating other real-world settings, such as passport control, remains unclear. This is due to the fact that alongside the YBT and the FICST, Stacchi *et al.* [42] employed two relatively short matching tasks which featured an equal number of match and mismatch trials (e.g. [17]). Conversely, face matching at passport control may be compounded by additional challenges. Related to the well-established effects of expectations on performance (e.g. [45]), these may include the number of passengers to be processed (i.e. [46,47]), the infrequent occurrence of identity mismatches (see [48,49]), and age-related variation in appearance [36,50]. At the same time, pairwise face matching tests also provide a limited proxy for the task of passport issuance and renewal, which entails a comparison between the applicant and an array of the most similar individuals returned by an algorithm who are already in possession of a passport (see [27]). It seems reasonable to suggest that these tasks do not fully approximate these challenges that are encountered in passport control or issuance. Thus, the capacity of the YBT and the FICST to predict performance in such tasks remains unclear.

In this study, we present an abridged version of the YBT featuring 10 items (the YBT-10), and explore possible associations between this task and several other tests that are intended to capture some of the difficulties faced by passport control and issuance personnel. These comprised the FICST [44], the long version of the Kent Face Matching Test (KFMT-long; [37]) and the 1-in-10 matching task [51]. Alongside these face matching tasks, observers also completed two measures of unfamiliar face recognition: the CFMT+ [28] and the Models Memory Test (MMT; [31]), both of which entail similar procedures, but vary in terms of stimulus material and task difficulty.

The aims of this study were threefold. First, in the light of the previously reported high difficulty of the YBT, whereby on average only 9 out of 35 identities were correctly matched [42], we sought to standardize a 10-item version to create a more economical tool for assessing unfamiliar face matching across age-related changes in appearance (see also [36,50]), for which we provide normative data.

Second, we aimed to establish the extent to which individual performance was associated across the six tests deployed here. This is especially important in the light of increasing reports of at best moderate correlations between individual tests of face processing (even within the same subprocess; e.g. [26,37,42,52]). The performance measures considered here were response accuracy (i.e. the proportion of correct responses) across all tasks, as well as sensitivity (i.e. $d'$), which represents a measure of combined performance across hits and false alarms, and considers response bias [53]. Given that a key

requirement of passport officers is to detect both matches *as well as* mismatches, we used a performance measure here that combines both match and mismatch accuracy into a single performance measure.

The final aim was to examine the similarity of performance profiles demonstrated by the top and bottom 5% of performers for each test, with a specific focus on establishing the degree of convergence between the top 5% of performers (see also [42]). Given recent evidence that some, but not all, super-recognizers are also super-matchers [31], we reasoned that different patterns of performance could emerge. For instance, top performers in face matching tasks may also exhibit superior performance in recognition tasks, and vice versa, but could just as likely exhibit distinct performance profiles across the different subprocesses and tests of face cognition (i.e. face perception and face recognition). Conversely, the bottom 5% of performers identified by a given test were anticipated to exhibit a more consistently poor level of performance that was not constrained to any one specific test or subprocess. Finally, in addition to exploring the degree to which individual extreme performance achieved in a specific test generalizes to another, we investigated the similarity among performance profiles when observers were ranked based on their *overall* performance across all tests.

# 2. Method

## 2.1. Participants

The sample for this study comprised 146 participants (67 males, 79 females) with a mean age of 30 years (s.d. = 12.91). This cohort comprised students from the University of Fribourg who participated in exchange for course credit, as well as non-student participants personally known to the experimenters (i.e. friends and family members). Each subject provided written consent to participate. All procedures were approved by the local Ethics Committee at the University of Fribourg.

An additional independent sample of 27 participants (11 females, 16 males; mean age: 37, s.d. = 12.28) was tested to investigate whether the YBT-10 represents a stable measure of face matching (for similar approaches, see [37]). Due to constraints related to in-person testing of human subjects related to the global pandemics, these observers completed two sessions of an online version of the YBT-10 (mean inter-session interval: 9 days, s.d. = 4; range: 6–20).

## 2.2. Procedure and tests

Of the six tests employed for this study, four measured face perception via *matching* performance, and comprised the YBT-10, a novel short version of the original 40 item YBT [42,43], the FICST [42,44], the KFMT-long [37] and the 1-in-10 task [30,51]. The remaining two tests measured face *recognition* ability via *memory* performance, and included the CFMT+ [28] and the MMT [31]. Examples for each test are provided in figure 1.

The YBT-10 and the FICST were administered in paper-based form, whereas the remaining four tests were computer-based, and were administered using *PsychoPy* (see [54]). To control for order effects, the sequence in which these tests were administered was randomized across observers. In addition, responses were always self-paced, and no performance-related feedback was provided at any point during the study. Finally, the computer-based tasks were administered across several different machines which varied in screen size, from 13.3 inches in width to 14 inches in width. Likewise, while the screen dimensions varied across testing devices, the size of stimuli onscreen was adjusted for each device to ensure that presentation size was consistent across participants, and was comparable to those used in previous studies (e.g. [31,37,42]). Below, each test employed is described in detail.

### 2.2.1. The 10-Item Yearbook Test

This paper-based task represents a shortened version of the recently standardized YBT [42], which was originally reported as a test of personally familiar face recognition after 25 years [43]. Previous findings indicate that observers unfamiliar with the depicted identities correctly match eight to nine items on the full version of the test (35 analysable items; see [42]). Here, we selected 10 items (5 male and 5 female target identities) that were correctly identified by approximately 30% of the previously tested observers [42]. These are presented on two different A4 pages (grouped by gender; completed in the same order: male, female), which each consist of a left column showing five target identities,

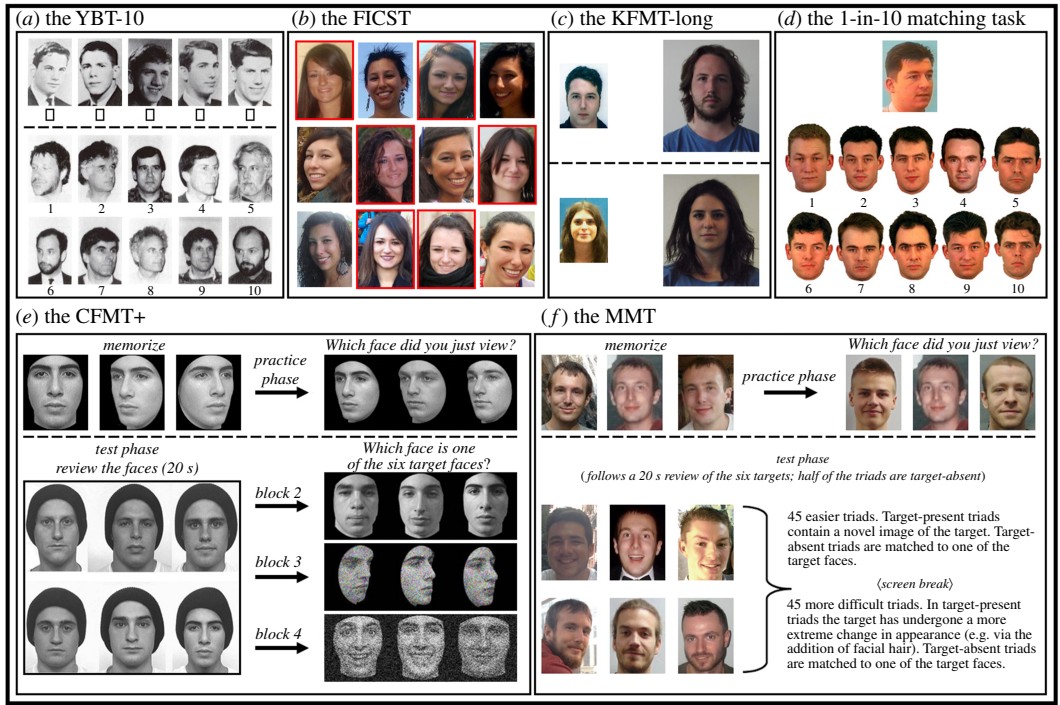

**Figure 1.** Example stimuli and procedures from the six tests that were administered in this study. Analogies of the (*a*) YBT-10 [43] and (*b*) FICST [44] (for illustrative purposes, the YBT-10 is reformatted, and the FICST is demonstrated using different identities). In (*b*), the two identities can be distinguished here based on whether they are marked with a red border. (*c*) Example match (top) and mismatch (bottom) stimuli from the KFMT-long [37] and (*d*) represents a trial from the 1-in-10 task [51]. Also depicted are stimuli and procedures employed in the (*e*) CFMT+ [28] and (*f*) the MMT [31].

alongside a 10-face array containing an older version of each target and five distractors (i.e. non-target identities; all of comparable age). To facilitate this task, participants submitted their responses by recording the corresponding number for each identity in the left column. All images were presented at a size of 220 × 350 mm. The paper-based version can be requested via email.

The second cohort of 27 observers completed a digitized version of the YBT-10, implemented as a single-page Web application. A calibration was implemented to ensure that on-screen stimulus size was identical to the physical stimulus size of the paper-based version. To accommodate for the restricted height on computer screens, the stimuli were laid out in a landscape arrangement (cf. figure 1*a*), as opposed to portrait the arrangement used on an (A4 page) paper-based analogue version of the test. Access via mobile devices (phones and tablets) was prevented.

### 2.2.2. The Facial Identity Card Sorting Task

Following Jenkins *et al.*'s [44] procedure, stimuli for this task comprise 20 face photographs of one unfamiliar Dutch celebrity that are intermixed with 20 face photographs of another Dutch celebrity of similar appearance and with whom observers were again unfamiliar, as standardized recently [42]. Observers were presented with these 40 photographs and were instructed to sort them into groups of as many identities as they believed were present. Critically, observers were not informed of the number of identities that were present across the 40 images. Consequently, a perfect score in this task would correspond to two groups of 20 cards, with each grouping containing images of only one identity. As in the YBT-10, this task was self-paced, and the time taken to complete this task was not recorded.

### 2.2.3. The Kent Face Matching Test—long form (KFMT-long)

The long version of the KFMT [37] consists of 220 pairs of faces, of which 200 depict the same person (match trials), and 20 depict different identities (mismatch trials). Each stimulus features two face photographs positioned on a white background. The right image comprises a digital face photograph that was collected in a laboratory, and which depicts the subject against a plain white background

while bearing a neutral expression and frontal pose. Conversely, the left image in each pair comprises a student ID photograph which was obtained with permission for each subject, and which was not constrained in terms of expression, pose or lighting. Importantly, on identity match trials, these images were taken several months prior to the controlled laboratory photograph, and thus represent an important source of variability (see also [50]). To ensure the even distribution of match and mismatch trials throughout the task, these were divided into four blocks of 55 face-pair stimuli (50 matches, 5 mismatches). Observers were instructed at the beginning of the task that mismatches would occur less frequently than match trials, but were not informed of the exact ratio. Responses were submitted via the 's' and 'd' keys on the keyboard, which indicated *same* and *different* identities, respectively. No breaks were administered between blocks.

### 2.2.4. The 1-in-10 matching task

This task comprises 80 facial arrays presented in colour that were constructed using the face images from the UK Home Office Police Information Technology Office database ([51]; see also [30]). Each trial features one target image that was extracted from video footage and presented in a three-quarter pose, positioned above a line-up of 10 frontally oriented faces on a plain white background. On target-present trials, a different image of the person depicted above the line-up is also present within the line-up. Conversely, on target-absent trials, the line-up is composed entirely of distractor faces. Observers were presented with 40 target-present and 40 target-absent trials in a random order, and were instructed to indicate the matching probe by pressing the corresponding numbered key on the keyboard (with 0 corresponding to 10), or the 'space' key if they believed that the target was absent from the line-up. In addition, observers were not informed that target-present and target-absent trials would be occurring with equal frequency.

### 2.2.5. The Cambridge Face Memory Test+

The long version of the CFMT comprises 102 trials that are divided across four blocks (see [28]), of which the first three consist of the 72 trials that make up the original CFMT (see [55]). In the first block, observers were instructed to study three concurrent images of the same target face across different orientations (frontal, mid-profile right and mid-profile left) for 3 s, and were then tasked with identifying the target from within a three-face array that featured the target face alongside two distractors. In the second block, observers studied six concurrently presented male faces for 20 s, and were then presented with a series of three-face arrays containing one of the previously studied faces alongside two distractors. For each array, observers were required to indicate which of the three faces featured in the study phase. The third block was procedurally identical to the second block, except with the addition of Gaussian noise to face images, so that texture could not be used to aid recognition. The final block of 30 trials further increased the difficulty of this task by introducing extra noise, as well as changes in pose and expression, and by removing external facial features [28].

### 2.2.6. The Models Memory Test

In the MMT [31], observers are required to encode and subsequently recognize faces of unfamiliar male models. During the first phase of this task, observers completed 18 trials, which entailed studying three photographs of the same target person that were presented sequentially for 3 s each. Observers then viewed an array of three face photographs, and indicated which of these was the target. This first block was treated as a practice phase, and did not contribute towards the final score. In the second phase of the task, observers viewed a concurrent array of six faces belonging to different identities, which were displayed for 20 s. This was followed by 90 trials divided into two sections of varying difficulty: target-present trials, consisting of a different image of one of the targets alongside two similar-looking distractors, and an equal number of target-absent trials. Target-present trials within the first section featured photographs of the target that were similar in terms of general appearance. Conversely, target-present trials in the final section incorporated a greater degree of within-person variability, such as changes in viewpoint, facial hair and paraphernalia. Finally, to enhance the difficulty of this task further, target-absent trials featured one identity that was similar in appearance to one of the six targets that were studied.

# 3. Analysis

## 3.1. Data preparation

For the YBT-10, raw scores were compiled and analysed; these scores range on a scale from 0 to 10. Conversely, computing a performance measure for the FICST is less clear-cut. In this task, observers are required to group 20 images of one person that are intermixed with 20 images of somebody else. Thus, to achieve a perfect score in this task, observers must sort images into two groups of 20, with each group containing images of a single identity. Consequently, there are two possible types of error that must be considered when calculating performance in the FICST: *distinction* errors and *conflation* errors. Distinction errors occur when observers create too many groups (i.e. greater than 2), and thus falsely perceive images of the same person as distinct identities. Conflation errors, on the other hand, occur when observers conflate images of different people into the same grouping. Following our recent procedure [42], to create a composite score that considers both types of error, we added the number of groups to the number of conflated identities, and then subtracted two from this total. This means that perfect performance in this task would correspond to a score of zero (i.e. [2 groupings + 0 conflation errors] − 2), with higher scores indicating worse performance.

For the computer-based tests, correct responses were converted into percentage accuracy scores. For the KFMT, each observer's overall accuracy was calculated based on an average of match and mismatch accuracy. In addition, for the KFMT, the 1-in-10 task, and the MMT, overall sensitivity ($d'$) was also calculated to provide a measure of overall performance that can account for bias in responding. The data for all experiments can be downloaded from the Open Science Framework (http://dx.doi.org/10.17605/OSF.IO/QA4JT); none of the experiments were preregistered. Additional measures and analyses complementing those reported below can be found in the electronic supplemental material.

## 3.2. Computation of individual observers' ranks per tests and overall rank across all tests

As we were interested in examining individual differences within the top and bottom proportions of our sample, as a secondary step to our analysis, all data were converted to z-scores, and were subsequently ranked for each test (with inverse rank scoring for the FICST). We next summed each individual's rank for each individual test to compute an overall rank which enabled us to identify the top and bottom observers across the entire sample and collection of tests.

## 3.3. Data analyses

To control for multiple correlations, *p*-values were adjusted throughout using Holm's sequential Bonferroni correction (but for alternative approaches, see [33,41,56]; see also [57]). All correlations were computed using the Spearman's rank method. The Kuder–Richardson formula 20 (KR20) was used to measure reliability for the four previously established tests: KFMT-long, CFMT+, MMT and 1-in-10. For the YBT-10, split-half reliability was computed for the sample reported here ($n = 146$) using the Spearman–Brown prophecy formula [58]. Test–retest reliability was estimated based on data from an independent cohort of observers ($n = 27$), who completed a recently implemented digitized version of the YBT-10 twice (see electronic supplemental material). In the light of its procedure, reliability cannot be estimated for the FICST.

# 4. Results

## 4.1. Analysis 1: summary statistics and tests' consistency

Cross-subject performance means and medians calculated for each test are provided in table 1, and figure 2 provides z-standardized performance distributions for all tests.

### 4.1.1. Performance measured for the YBT-10

For the newly assembled YBT-10 task, the 146 observers correctly matched 3.71 (s.d. = 1.99) out of 10 identities (i.e. 37%), indicating that this tool is somewhat easier than the recently reported long version of this task (YBT; mean of 8.8 out of 35 correctly matched, i.e. 25%; [42]). The split-half (odd–even across trials grouped by sex) correlation across items was 0.45, and 0.62 with the Spearman–Brown

**Table 1.** Cross-subject performance means and medians for each test employed. The final column indicates for each test the cut-off score at which individuals of the sample tested perform within the top 5% bracket; standard deviations are provided in parentheses.

| test | mean (s.d.) | median | min | max | top 5% cut-off |
|---|---|---|---|---|---|
| YBT-10 | 3.71 (1.99) | 3.5 | 0 | 9 | 7 |
| FICST | | | | | |
| total score | 7.53 (5.17) | 7 | 0 | 24 | 0 |
| number of groups | 7.56 (4.62) | 7 | 1 | 25 | |
| number of errors | 1.97 (3.62) | 1 | 0 | 20 | |
| KFMT-long | | | | | |
| overall % | 68.21 (7.88) | 68.75 | 47.75 | 86.75 | 80% |
| match % | 73.83 (16.62) | 75.50 | 16.00 | 99.50 | |
| mismatch % | 62.60 (22.62) | 65 | 10 | 100 | |
| $d'$ | 1.12 (0.46) | 1.14 | −0.22 | 2.24 | |
| 1-in-10 | | | | | |
| overall % | 58.19 (15.90) | 57.5 | 21.25 | 91.25 | 85% |
| hits % | 69.25 (15.69) | 72.50 | 20.00 | 97.50 | |
| correct rejections % | 47.14 (26.32) | 42.50 | 0.00 | 97.50 | |
| $d'$ | 0.49 (0.98) | 0.48 | −1.83 | 2.80 | |
| CFMT+ | | | | | |
| accuracy % | 63.80 (11.35) | 64.71 | 29.41 | 88.24 | 83% |
| raw score | 65.08 (11.58) | 66 | 30 | 90 | 85 |
| MMT | | | | | |
| overall % | 50.27 (15.79) | 51.67 | 16.67 | 81.11 | 74% |
| hits % | 49.83 (16.88) | 51.11 | 13.33 | 84.44 | |
| correct rejections % | 50.72 (26.75) | 53.33 | 0.00 | 100 | |
| $d'$ | 0.01 (0.95) | 0.09 | −2.08 | 2.24 | |

adjustment. Given the items' arrangement (five male targets, followed by five female targets), we explored the correct responses as a function of item sex. This revealed that observers scored comparably when all items of a given sex were considered ($M_{male}$ = 1.87, s.d. = 1.21; $M_{female}$ = 1.84, s.d. = 1.39). However, inspection of performance for individual items revealed differential difficulty across individual male and female items. Two male items were matched correctly by 23% of observers, while the remaining three were matched correctly by 44%, 48% and 50%, respectively. For female items, observers' performance across items varied comparatively less, and ranged from 28% to 43%, with two items correctly matched by 38%, and one item by 37% of observers. Split-half reliability computed based on performance for the first and last five items, respectively, yielded $r_{predicted}$ = 0.28.

To further determine the reliability of the YBT-10, an independent sample completed an online version of this test, at two separate time points (T1, T2). The mean accuracy at T1 and T2 was 3.70 (s.d. = 2.03) and 3.78 (s.d. = 2.17), respectively. These scores converge with those observed in our main sample of 146 subjects, for which average accuracy was 3.71 (s.d. = 1.99). Accuracy across all 10 items correlated between T1 and T2 ($r$ = 0.44, $p$ = 0.02). Split-half reliability computed for male versus female items at T1 yielded $r_{predicted}$ = 0.24, and $r_{predicted}$ = 0.36 at T2. Correlating performance across sessions (T1 versus T2) as a function of stimulus sex confirmed that male items were associated with more reliable performance compared to female ones ($r_{predicted}$ = 0.79 versus $r_{predicted}$ = 0.29, respectively).

### 4.1.2. Performance measured for established tests of face cognition

Similarly, scores obtained in the FICST ($M$ = 7.53; s.d. = 5.17) converged with typical behaviour reported previously in this task (see [42,44]). Overall mean accuracy in the KFMT-long was 68% (s.d. = 7.88%),

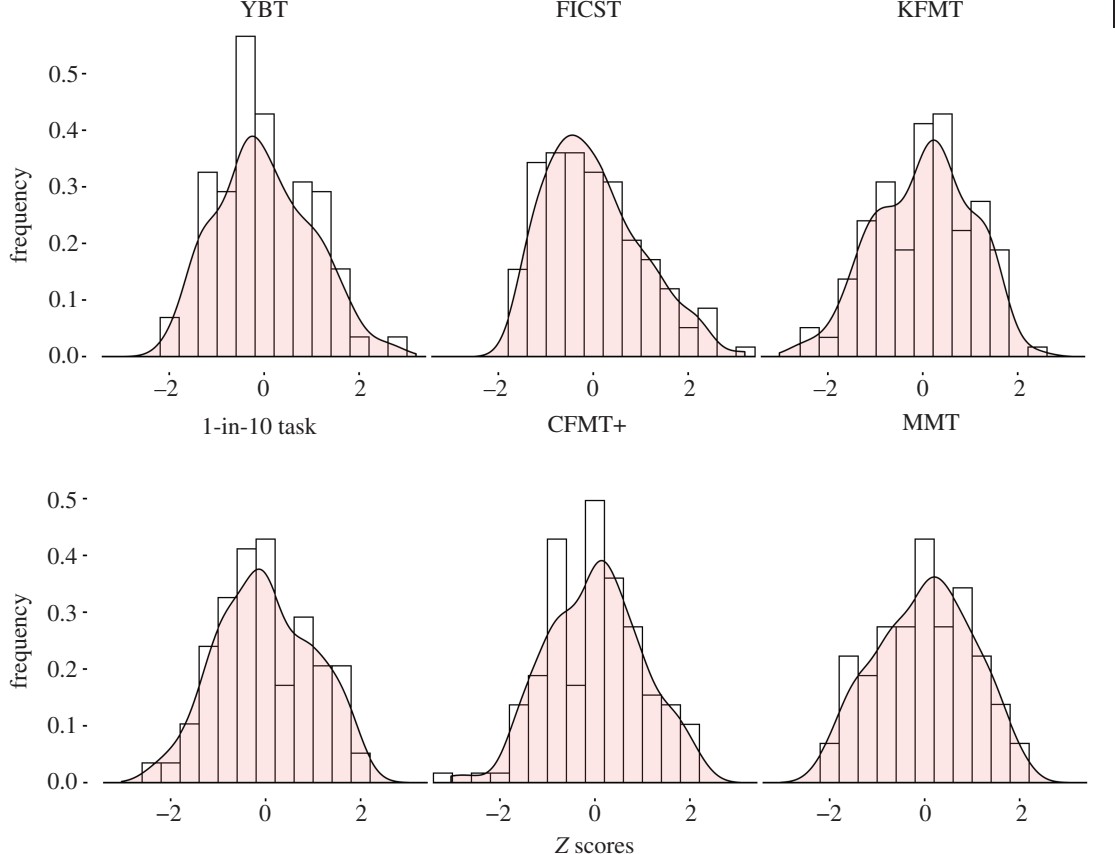

**Figure 2.** Distribution of standardized performance observed for the six administered tests. Displayed here are frequencies of (z-scored) accuracy scores per test.

with accuracy rates of 74% (s.d. = 16.62%) and 63% (s.d. = 22.62%) on its match and mismatch subcomponents, respectively, which aligns with previously reported normative accuracy in this task (see [37]). Likewise, $d'$ in this task was 1.12, and *criterion* was −0.17. One-sample $t$-tests revealed that *criterion* was below zero, implying a bias to classify trials as identity matches, $t(145) = 3.53$, $p < 0.01$; the KR20 coefficient for this task was high, at 0.96. For the 1-in-10 task, the overall mean accuracy was 58% (s.d. = 15.90%), with 69% (s.d. = 15.69%) and 47% (s.d. = 26.32%) accuracy on target-present and target-absent trials, respectively. The mean $d'$ in this task was 0.49, and the KR20 coefficient was 0.90. Finally, percentage accuracy in the CFMT+ ($M = 64$%; s.d. = 11.35%) and the MMT ($M = 50$%; s.d. = 15.79%) converged with recently published norms for these tasks (e.g. [28,31]), and $d'$ on the MMT was 0.01. Likewise, KR20 coefficients indicate high reliability for both tasks (CFMT+: 0.87; MMT: 0.90).

## 4.2. Analysis 2: correlations between tests

Figure 3 visualizes the Spearman's correlations of performance exhibited across tests. Following the suggestions of two anonymous reviewers, the measures subject to correlations were observers' accuracy for the YBT-10 and CFMT+, FICST score; for the three tests that incorporated target-absent trials (KFMT-long, 1-in-10 and MMT), to consider response biases, $d'$ was considered in the correlation analyses (see electronic supplemental material for correlations performed on accuracy scores). All correlations were significant following Holm's sequential Bonferroni procedure [59], suggesting high degrees of overlap both within and between subprocesses of interest (i.e. face perception and face recognition).

## 4.3. Analysis 3: inspection of observers located at the extremes of the performance distributions

In addition to the correlations reported above, we explored the relationship between test scores at the level of individual observers. This was conducted in two steps: the top and bottom 5% of all observers were identified (i) *independently per test* and (ii) based on their *overall performance* across all tests.

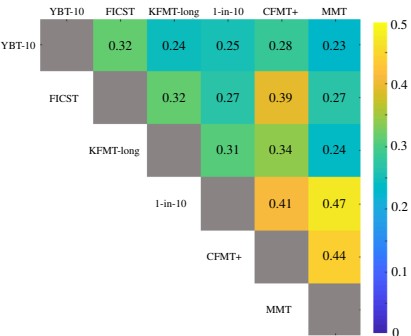

**Figure 3.** Visualization of Spearman correlations of performance measures for the six tests employed. Measures considered were accuracy in the YBT-10 and CFMT+, FICST score and $d'$ in the KFMT-long, 1-in-10 and MMT, respectively. Cell contents indicate individual $r$ values; all (Holm's Bonferroni corrected) correlations reached significance.

**Table 2.** Mean performances for the (a) top 5% and (b) bottom 5% of performers identified independently per test. Each column represents the top or bottom 5% of performers for the specified test, respectively, and each row indicates how well this subsample performed on the additional tests that were employed in this study. Reference means for each subset are provided in italics.

| (a) | top 5% of performers | | | | | | | | | | | | |
|---|---|---|---|---|---|---|---|---|---|---|---|---|---|
| | accuracy | | | | | | | z-scores | | | | | |
| | YBT-10 | FICST | KFMT | 1-in-10 | CFMT+ | MMT | mean (s.d.) $n = 146$ | YBT-10 | FICST | KFMT | 1-in-10 | CFMT+ | MMT |
| YBT-10 | *7.86* | 5.43 | 4.00 | 4.43 | 5.71 | 4.29 | 3.71 (1.99) | *2.09* | 0.80 | 0.15 | 0.36 | 1.01 | 0.29 |
| FICST | 5.14 | *0.00* | 6.86 | 3.71 | 3.86 | 3.86 | 7.53 (5.17) | 0.79 | *1.46* | 0.13 | 0.74 | 0.71 | 0.71 |
| KFMT | 69.75 | 74.89 | *81.71* | 73.00 | 66.54 | 74.32 | 68.21 (7.88) | 0.34 | 0.85 | *1.71* | 0.61 | −0.21 | 0.78 |
| 1-in-10 | 65.36 | 65.89 | 52.14 | *87.14* | 68.21 | 72.86 | 58.19 (15.90) | 0.82 | 0.59 | −0.38 | *1.82* | 0.63 | 0.92 |
| CFMT+ | 72.69 | 69.47 | 67.79 | 72.97 | *85.85* | 69.61 | 63.80 (11.35) | 1.01 | 0.47 | 0.35 | 0.81 | *1.94* | 0.51 |
| MMT | 53.65 | 58.10 | 52.06 | 62.54 | 66.51 | *78.73* | 50.27 (15.79) | 0.65 | 0.26 | 0.11 | 0.78 | 1.03 | *1.80* |
| (b) | bottom 5% of performers | | | | | | | | | | | | |
| | accuracy | | | | | | | z-scores | | | | | |
| | YBT-10 | FICST | KFMT | 1-in-10 | CFMT+ | MMT | mean (s.d.) $n = 146$ | YBT-10 | FICST | KFMT | 1-in-10 | CFMT+ | MMT |
| YBT-10 | *0.43* | 2.71 | 2.57 | 1.43 | 2.86 | 3.29 | 3.71 (1.99) | *−1.65* | −0.50 | −0.57 | −1.15 | −0.43 | −0.21 |
| FICST | 7.14 | *19.57* | 13.14 | 14.00 | 11.14 | 9.00 | 7.53 (5.17) | 0.07 | *−2.33* | −1.09 | −1.25 | −0.70 | −0.29 |
| KFMT | 64.54 | 67.11 | *51.79* | 65.86 | 61.75 | 67.32 | 68.21 (7.88) | −0.47 | −0.14 | *−2.09* | −0.30 | −0.82 | −0.11 |
| 1-in-10 | 69.11 | 55.36 | 43.75 | *26.61* | 37.14 | 40.36 | 58.19 (15.90) | 0.69 | −0.18 | −0.91 | *−1.99* | −1.32 | −1.12 |
| CFMT+ | 61.76 | 53.50 | 61.62 | 51.12 | *41.18* | 51.82 | 63.80 (11.35) | −0.18 | −0.91 | −0.19 | −1.12 | *−1.99* | −1.06 |
| MMT | 48.10 | 45.71 | 45.56 | 36.51 | 33.17 | *20.16* | 50.27 (15.79) | −0.14 | −0.29 | −0.30 | −0.87 | −1.08 | *−1.91* |

### 4.3.1. Top and bottom 5% of observers identified independently per test

In line with Stacchi *et al.* [42], we next isolated the top and bottom 5% of the cohort tested here ($n = 7$) for each test (based on $z$-standardized accuracy scores) to examine how these individuals performed in relation to the overall sample mean. These were randomly selected in cases where more than seven observers achieved the highest possible score, such as in the FICST. Cross-subject means for these subsamples are depicted in table 2(a,b), respectively, for each test. From these data, it can be observed that the top 5% of observers in each test exceeded the sample mean by at least one s.d. However, there was limited evidence of overlap between tests among higher performers, whereby only the top performers in the YBT-10 also exceeded the sample mean of a different test by at least one s.d. (the CFMT+).

**Figure 4.** Visualization of observers' performance profiles after ranking based on their overall performance across all tests. Radar plots on the left demonstrate performance for individuals who ranked highest (top row) and lowest (bottom row) across all tests. Blue areas denote individual performance, and the grey areas denote the mean performance across the entire sample. The PSM on the right characterizes the relationship between individual performance profiles for observers ranked according to their overall performance across all tests. The black highlight shows the area of the PSM for high and low performers, respectively.

Similarly, the data for the bottom 5% of observers for each test verify that for each measure, the average performance for these low performers was at least one s.d. below the sample mean. Observers who were in the lower accuracy range on KFMT were also poor in the FICST, while low scorers on the 1-in-10 were also in the bottom 5% bracket on the YBT-10 and the FICST. In addition, low performers on the CFMT+ also scored one s.d. below the sample mean in the FICST and the MMT, whereas lower-range accuracy in the MMT coincided with similarly poor performance in the 1-in-10 and the CFMT+. While the majority of observers only fell into one low-performing bracket, nine observers who were among the bottom 5% in one test were also among the low performers in at least one other test. Of these, one observer was among the bottom 5% in three tests (CFMT+ (30%), 1-in-10 (21%) and MMT (24%)). In addition, one observer was counted among the bottom 5% of performers across four tests (CFMT+ (45%), 1-in-10 (21%), KFMT (48%) and FICST (24)).

### 4.3.2. Overall top and bottom 5% of observers identified across all tests

Finally, we inspected the patterns of performance exhibited by observers who were the top and bottom 5% based on their rank sum *across all tests* employed here. The radar plots depicted in figure 4*a* display each of the seven overall high and seven overall low performers, respectively. This visualization demonstrates the variability at the extremes of the performance distribution across the tests of face matching and recognition used here. The performance similarity matrix (PSM) in figure 4*b* describes this relationship not only among the overall top and bottom performers, but across all *the entire sample tested*. As evident from this illustration, no rank-dependent clustering emerged.

# 5. Discussion

In this study, observers completed six tests of unfamiliar face processing that were assembled as a potential battery of tests for the selection of personnel in applied settings. Four of these tests (the YBT-10, FICST, 1-in-10 and KFMT-long) measured observers' ability to match images of unfamiliar faces (i.e. face *perception*), whereas the remaining two tests (the CFMT+ and MMT) assessed observers' capacity for encoding, learning and subsequently recognizing faces (i.e. face *recognition*). Testing a large cohort of individuals with all six measures, we sought to explore the performance relationship between tests and observers, with a particular focus on the three questions discussed below.

## 5.1. Development of a challenging screening test of face matching: the YBT-10

One of our primary aims was to develop and standardize the YBT-10, which comprises a shorter version of the original YBT [42,43], featuring only 10 to-be-matched target identities. On average, in this new screening version, our participants correctly matched three to four items (i.e. 37%), which aligns with recent work, in which observers correctly matched 25% of the target identities on the complete version of the YBT [42].

Investigating the YBT-10's internal and test–retest reliability in independent samples of normal observers revealed differences related to item sex. Specifically, lower reliability was found for female items. This was attributed to a selection of male items being more frequently correctly matched, i.e. being less difficult than their female counterparts.

Despite the item-sex-dependent differential reliability observed, we suggest that the YBT-10 should not be dismissed as a potentially challenging screening test to detect superior performance, for two reasons at least. First, reliability aside, it is unlikely that a normal observer would achieve high or excellent performance by chance. In fact, not a single observer tested achieved the maximum score of 10 (out of 146 observers, only four scored eight or nine correct). Second, it is not sensible to determine the reliability of a test designed to screen for *superior* skill among a group of *normal* observers. It is plausible that higher test reliability would be observed if performance measured from individuals with superior face cognition skill, i.e. so-called super-recognizers [23,28], were subject to analysis.

We, therefore, propose that the YBT-10 complements the original version of the test, by providing a streamlined measure of performance in this task, which is sufficiently challenging so as to amplify individual differences in face processing at the higher end of the ability range. Further studies are required to establish whether this 10-item screening test and the longer original version can reliably distinguish between super-recognizers identified with other measures, and normal control observers.

## 5.2. Association between performances across tests of face matching and recognition at the group level

Our second aim was to explore possible associations between the six tests deployed here (figure 3). Converging with Stacchi *et al.*'s [42] findings with the original long version of the YBT, here the YBT-10 also correlated with the FICST and the CFMT+. Moreover, here performance in the YBT-10 was also associated with performance in the KFMT-long and the 1-in-10 task. While the KFMT-long and the 1-in-10 also represent tasks of face matching, they vary in terms of procedure and difficulty. These correlations indicate that the YBT-10 relies on mechanisms similar to those recruited for other matching and memory tasks.

Alongside the performance associations for the YBT-10, significant correlations were also observed between the KFMT-long, the FICST and the 1-in-10 task. These relationships may indicate that all four face matching tasks featured in this study can be solved through similar strategies. While this would not necessarily be surprising, it is also not a foregone conclusion, given that these tasks also differ in some aspects. Consider, for example, that while both the KFMT and the 1-in-10 task measure facial matching, these tasks differ in terms of the underlying probabilities, and hence task difficulty. In the KFMT-long, observers decide whether *pairs* of faces show the same person or different people. In the 1-in-10 task, observers must instead locate a target from within an array of 10 probe faces, while also keeping in mind that the target might not be present in the probe array. Finally, in the FICST observers group photographs of unfamiliar people via a series of photo-to-photo comparisons, where performance in this task is also susceptible to factors such as participants' belief about the task [60] and their own ability [61]. Despite these procedural differences between the matching tasks that were employed here, however, the correlations observed here imply that performance in these tasks is related, probably due to application of a similar mechanism or strategy.

Converging with recent work (e.g. [31]), the CFMT+ and the MMT correlated positively. In addition, performance on these face memory tasks was also associated with the face matching tasks, although these correlations varied in strength. The KFMT-long, for instance, correlated with the CFMT+ (see also [37,41]), as well as the MMT. These findings may be taken to support studies reporting correlational group analyses, suggesting that engaging in different tasks may involve the recruitment of both common, and independent mechanisms (e.g. [33,39,42]).

It is worth noting here that there was evidence of a response bias on the 1-in-10 task which was characterized by a tendency to select a face within the probe array even when the target was not present. Currently, it is not clear as to why observers were reluctant to submit a 'target-absent' response. One possibility, however, is that this bias was carried over from the other tests which do not feature target-absent trials (YBT-10, CFMT+, FICST), as well as the KFMT-long, in which mismatch trials occur infrequently. Note, however, that the correlations performed (figure 3) considering $d'$ as a performance measure for tests that included target-absent trials. Further work is required to systematically determine the effect of incorporating only target-present trials (e.g. CFMT+), as opposed to also including (different proportions of) target-absent trials. Another interesting question raised by an anonymous reviewer concerns the issue of whether the size of our sample is sufficient, which we believe is the case. Our sample exceeds that of many recent correlational studies (cf. [34] ($n = 114$); [37] ($n = 50$); [41] ($n = 70$); [33] ($n = 103$); [11] ($n = 40$)), and converging with previous findings, we report correlations between different tests of face matching [34,37], as well as between face matching and face memory [11,33,37]. Of course, it is possible that larger samples could give rise to stronger correlations (e.g. [62]).

## 5.3. Focus on the individual: high performance in one test does not (necessarily) predict that exhibited for another test

Having demonstrated correlations between independent tests of face perception and recognition at the group level, our final aim was to explore differences in observers' performance profiles in greater detail. The rationale was that, while undoubtedly meaningful, relationships between tests observed at the group level in large cohorts will mask variability that is important when decisions are required on an individual basis. For instance, personnel selection involves assessment of each individual's performance, where only a limited amount of information is available per individual, or only a small number of individuals are tested for selection [23]. To maximize the value of this limited amount of available information, it is critical to understand the relationships across subprocesses (e.g. superior

face perception, but not face recognition performance), as well as across tests of the same subprocess (e.g. all tests requiring face matching, or face memory, respectively).

Complementing the pairwise relationships assessed via the aforementioned correlational analyses (figure 3), we investigated how performance varied among the top and bottom 5% of observers for any given test, in relation to the other measures that were employed. Among high performers, we found limited evidence for generalization between tests. For instance, the mean accuracy for top performers in one test was frequently within one s.d. of the sample mean across other tests. In addition, no observers scored among the top 5% across all six tests; only one observer fell into this bracket across three tests (YBT-10, FICST and CFMT+). In line with recent evidence, the top performing individuals in the CFMT+ also showed little evidence of superior ability in the other tasks. This emphasizes that conventionally used cut-offs must be utilized *in combination* with other difficult measures of face processing [23,31,42,63]. There was similarly little evidence of overlap between measures among the bottom 5% of observers in each test. Although one observer was in the bottom range for four out of the six tasks, the majority of individuals' low performance levels were confined to a specific task. This indicates that it is possible for one to be deficient in one measure of face matching, such as the YBT-10, but not the KFMT-long, despite both tapping into face perception and requiring observers to match identity age-invariantly.

The most parsimonious account for the lack of overlap observed among both high and low performers is that different tasks will be solved using strategies that vary according to their specific task requirements, and that generalize more or less proficiently for solving other tests. For instance, someone may be severely impaired at matching identity in a viewpoint-invariant manner, without being concurrently impaired at processing isolated features, as exhibited by individuals with prosopagnosia [64,65]. This would lead to a specific performance profile across different tasks. Whether or not this is associated with impairments will depend on the difficulty of a given task, as well as the performance measures considered (i.e. accuracy alone, versus in combination with response times). For example, others have reported individuals with developmental prosopagnosia to exhibit normal performance under optimized test conditions (see [66]). Ultimately, before tests are used to separate average- from high-performing individuals, their ability to detect low-performing or impaired individuals needs to be established.

## 5.4. Representing the similarity of multi-dimensional performance profiles within and between observers

In the prior steps, we characterized the relationship between tests on the group level (figure 3), and through exploration of top performers identified based on a single test (table 2). Our final aim was to explore the similarity between observers, when considering their *overall performance* across all tests administered. To this end, after ranking individuals according to their overall performance, we computed the performance similarity between observers' profiles across all tests (figure 4). Here, we *zoomed in* on the extremes of the overall performance distribution, which is to say, we focused on individuals identified as overall high or low performers within the sample tested. Aligning with the previous observations of at best modest generalization from one test to another, the performance similarity across observers ranked by their overall performance did not systematically vary. This indicates that, rather than individuals at the extremes exhibiting more homogeneous performance profiles across the subprocesses and tests of face cognition, individuals generally exhibit highly distinct profiles irrespective of their overall performance.

## 5.5. Honouring individual differences in applied settings

Our observations have important implications for the deployment of (batteries of) tests for personnel selection. First, using a *single test* makes sense only if it accurately maps the task of interest and its real-world procedural constraints. Even if the test has been developed to meet this common-sense requirement, it is still necessary to empirically determine whether performance on this test indeed transfers into the professional domain (cf. [23]). The alternative solution would seem, therefore, to be adopting a *battery of tests* to assess one skill with various measures, or to assess complementary skills. This approach bears the potential for abundant information, but also a high degree of variability across observations. Consequently, based on the current findings, personnel selection based on a battery of tests should be conducted not simply using any given selection of tests. Rather, these

individual tests should systematically differ in a way that is meaningful for the real-world area of deployment, and depending on whether the goal is to identify individuals with very similar, or distinct and complementary skills [23,35].

## 5.6. Conclusion and future outlook

In this study, we developed the YBT-10 as a novel tool to economically and conveniently assess individual face matching ability, and related observers' performance to that exhibited for previously reported tests of face cognition. Adding to tests developed to capture the challenges associated with matching and recognizing faces in the real world (e.g. [17,31,37,42,67]), we believe that the YBT-10 provides a worthwhile contribution to the literature: it is a difficult performance measure that can amplify differences between high-performing observers, and it is easily administered.

Extending previous work, our detailed investigation of the relationships between and within individual performance profiles across six tests completed by a large group of observers revealed individual differences between tests probing the same subprocess. For instance, some individuals proficiently matched pairs of faces, but struggled with (some versions of) facial arrays. These findings extend work showing that not all super-recognizers are superior at face matching (e.g. [25,31]) as they demonstrate that low-performing individuals do not typically show evidence of generalization *between*, or even *within* the subprocesses that facilitate face matching and face recognition. Further investigations are required to determine whether the absence of generalization at both ends of the spectrum of a given subprocess is accounted for by procedural differences between measures, and/or intra-individual consistency [9,23,38,42,45,63]. Future studies involving multiple measures that vary systematically according to professionally relevant dimensions are needed to establish the isolated and combined predictive value of empirical tests for personnel selection.

Data accessibility. Accompanying data can be downloaded from the Open Science Framework (http://dx.doi.org/10.17605/OSF.IO/QA4JT).

Authors' contributions. L.S. and M.R. designed the study, supplied stimulus materials, implemented the experiments and oversaw data collection. M.C.F. and M.R. drafted the manuscript and analysed the data. All authors were involved in revising the manuscript, approved its final version and agree to be accountable for all aspects of the work in ensuring that questions related to the accuracy or integrity of any part of the work are appropriately investigated and resolved.

Competing interests. We declare we have no competing interests.

Funding. M.R. is supported by a Swiss National Science Foundation PRIMA (Promoting Women in Academia) grant no. PR00P1_179872.

Acknowledgements. We would like to thank Marine Ansermet, Daniela Barros Rodrigues, Chiara Buono, Quentin Bourqui, Lara Cattaneo, Julia Giovannini, Nicla Grandi, Anouk Labbé, Tess Quartenoud, Ludivine Riedweg, Lucia Schertenleib and Delphine Waeber, for assisting with data collection on this project. We also thank two anonymous reviewers for their constructive comments on a previous manuscript version.

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
