## [Reviewer comments · Royal Society Open Science]

Review History

RSOS-200233.R0 (Original submission)

Review form: Reviewer 1

Is the manuscript scientifically sound in its present form?

Yes

Are the interpretations and conclusions justified by the results?

Yes

Is the language acceptable?

Yes

Do you have any ethical concerns with this paper?

No

Have you any concerns about statistical analyses in this paper?

Yes

Recommendation?

Major revision is needed (please make suggestions in comments)

Comments to the Author(s)

This manuscript presents an analysis of a face processing battery. The authors examine the performance of over 140 individuals on multiple tests of face matching and face memory, to determine the level of consistency between and within different sub-processes of face recognition. Their analysis focuses on the group level associations, as well as associations at the extreme ends of the ability spectrum. While many of the tests show modest correlations, the findings indicate substantial heterogeneity in performance, even for high/low performers who are tested on tasks which (in theory) tap into similar processes.

Overall, I found the paper well-written. The tasks and analysis were described clearly, and the introduction and discussion addressed a number of relevant issues around the use of screening batteries in applied settings. I feel that the manuscript could be strengthened by considering the psychometric properties and different trial types within the tests in more detail, and considering some of the theoretical implications of their finding in addition to the more practical/policy-based implications.

Major comments:

1) Overall, I think the approach that the authors have taken in the results section is interesting and appropriate to their question. However, I believe that the reliability of the different tests (or even sub-components of the tests, see below) is an extremely important consideration, and the authors should (where possible) calculate or report this information. This is particularly important when the tests are being used to assess individual differences, and where correlations are being calculated. If tests have relatively low reliability, that can constrain the maximum correlation between tests – for example, if two tests have reliabilities of .86 and .40, the maximum correlation is around .59. If this is the case, a correlation of .35 is quite impressive! You can also use reliability to get attenuated correlations – this may provide a clearer estimate of the real relationship between the different measures. Bowles et al. (2009; doi:10.1080/02643290903343149) include a good example of these concepts applied to face recognition tasks.

While reliability estimates may be difficult for some of the tests in the battery (the sorting task is an interesting challenge), they should be fairly straightforward for most of the others. Including an estimate of reliability is also very important if the authors want to claim that their new, shortened Yearbook Test has appropriate psychometric properties for assessing individual differences.

2) I'm a bit confused as to why the IE measure was analysed for one measure but not any others. What is the justification for this? Presumably it isn't purely mathematical, since the IE doesn't meet the requirements outlined by Bruyer & Brysbaert (2011). While RT-based measures can provide an interesting additional perspective on face processing, and might be relevant for understanding different aspects of face cognition, I would suggest that the authors rethink their use of IE for the KFMT alone, and either examine correct RTs for all the tasks where that is appropriate/accessible, or remove RT-based measures from consideration entirely.

3) I am also concerned that performance in target-present/hit trials and target-absent/correct-rejection trials is being conflated in several tasks, particularly the 1-in-10 and MMT. There's a reasonable amount of evidence that performance on these trials tends to dissociate (e.g., Bate et al. 2019, DOI: 10.1002/acp.3525 for a recent example), and given the focus of the paper (i.e., examining consistency across different sub-processes of face processing) this seems an unusual omission. Even if/when the overall performance is of interest (and most of the time I believe the tasks should be broken down to separate the TA/TP trials – this has the potential to be far more informative at both the group and individual level), then the appropriate measure in these tasks is d' or A , which account for bias in responding.

4) The introduction and discussion would benefit from a more theoretical orientation. For example, the point that some tasks are more likely to index viewpoint-invariant processing whereas others can be solved using a more feature-based strategy is interesting and relevant to face processing models (and might be explored further using RT/IE-based measures).

Minor comments:

- 1) I found the use of “face recognition” as distinct from “face matching” quite confusing terminology (as typically when you’re matching a face on the basis of identity there’s an element of recognition involved as well). In the abstract, the authors refer to face memory vs face matching – this is much clearer, and I think it would be a good adjustment to apply throughout.
- 2) You mention that responses were always self-paced: were participants aware that RT was being monitored in the KFMT task? If not, this provides further justification for removing the IE measure from analysis.
- 3) Figure 1A (showing the YBT-10) doesn’t seem to align with the textual description of the task – it seems that the figure shows the to-be-matched faces on top, rather than on the left?
- 4) I’m pretty sure the MMT doesn’t ask people to remember 102 faces...? It’s difficult, but not impossible!
- 5) Given the sheer number of correlations being conducted (and the conservative nature of a straight Bonferroni correction), have the authors considered Holmes’ sequential Bonferroni procedure when correcting for multiple comparisons?
- 6) What were the correlations between IE for the KFMT and other measures? These should be reported in-text or included in the correlation table
- 7) I found Table 2 very hard to read and interpret. Perhaps the authors could reconfigure it to z-scores instead of raw scores? This would enable much easier comparisons between mean performance on different tests.
- 8) The authors mention that there were two poor performers whose performance was in the bottom 5% on multiple tests – similar information would be useful for the top performers as well (something along these lines is mentioned in the discussion but it should be reported in the results).
- 9) It’s difficult to say that the YBT-10 is “no less reliable” (p. 20, line 37) without calculating/reporting reliability. Certainly, the data supports convergent validity, which is great, but more work is needed to say that it’s equally reliable.
- 10) I am also curious as to why the KFMT-long and the YBT-10 did not correlate, given their general similarity. Have the authors investigated their proposal (re the length of the KFMT) by examining correlations for the first section of the KFMT only (assuming the sections are presented in stable order/are equivalent difficulty)?

Review form: Reviewer 2

Is the manuscript scientifically sound in its present form?

No

Are the interpretations and conclusions justified by the results?

No

Is the language acceptable?

Yes

Do you have any ethical concerns with this paper?

No

Have you any concerns about statistical analyses in this paper?

Yes

Recommendation?

Major revision is needed (please make suggestions in comments)

Comments to the Author(s)

This study reports individual differences data from six tests of face abilities. The goal of the study is threefold: provide normative data for one test, examine correlations between tests, and compare between-test performance of the top 5% and bottom 5% performers. The authors conclude that there is little relationship between performance across tests, and so they recommend the use of multiple tests for personnel selection.

The paper is well-written and the research question is important. I agree with the authors that we need to develop better tests of face processing abilities. Such tests would help us understand the mechanisms that contribute to the abilities and select forensic/security personnel with superior face recognition skills. I think this paper can make important contributions to the literature, but I have several major concerns:

1. The sample size of this study is too small for the authors to draw solid inferences. Modelling studies suggest a sample size of 250+ is needed to estimate a stable correlation between two typical measures in psychology (e.g., Schonbrodt & Perrugini, 2013). That number is likely to increase for this study since it compares multiple correlations across six measures. The authors should increase the current sample size by a considerable extent. It may even be useful to do a power analysis based on predictions about specific patterns of results, for example a strong association between two matching tests and a weak(er) association between a matching test and a recognition test.
2. Analysis 2 is incomplete because it doesn't test whether the correlations are statistically different from each other. For example, the claim that CFMT+ and MMT+ are different because CFMT+ correlates with YBT-10 but MMT does not is based on the mistaken assumption that the difference between a significant correlation (e.g., $r = 0.28$ for CFMT+ and YBT-10) and a non-significant correlation (e.g., $r = 0.23$ for MMT and YBT-10) must be significant. To support this claim the authors have to test whether $r=0.28$ and $r=0.23$ are statistically different, typically done using the Fisher's r to z transform.
3. How reliable are the six tests? It's difficult to make sense of individual-level analysis without knowing how much measurement noise the data contains. Maybe the tests are highly reliable, and so the weak generalisation across tests means that the tests are truly capturing different aspects of face processing abilities. But maybe the tests are noisy enough to allow meaningful comparisons of the 5% top/bottom performers. At a minimum, the authors could compute some sort of reliability indices (e.g., Cronbach's alpha) and incorporate them into their analyses and inferences.
4. Very few studies report the test-retest reliability of standardised face recognition tests (e.g., CFMT in Wilmer et al 2010). Test-retest reliability is necessary if researchers are seriously interested in developing valid and consistent measures of face abilities, especially if we want to use them in high-stake forensic/security settings. I highly encourage the authors to rerun the six tests with the same participants (or a subset of them) and compute their test-retest reliabilities. This would be a huge contribution to the field. A good model is the paper by Goodbourn et al (2012), which tests 1000+ participants with four measures of magnocellular processing and retests 100+ of them one week later.

Decision letter (RSOS-200233.R0)

14-Apr-2020

Dear Dr Ramon,

The editors assigned to your paper ("Differences between and within individuals, and sub-processes of face cognition – Implications for theory, research, and personnel selection") have now received comments from reviewers. We would like you to revise your paper in accordance with the referee and Associate Editor suggestions which can be found below (not including confidential reports to the Editor). Please note this decision does not guarantee eventual acceptance.

Please submit a copy of your revised paper before 07-May-2020. Please note that the revision deadline will expire at 00.00am on this date. If we do not hear from you within this time then it will be assumed that the paper has been withdrawn. In exceptional circumstances, extensions may be possible if agreed with the Editorial Office in advance. We do not allow multiple rounds of revision so we urge you to make every effort to fully address all of the comments at this stage. If deemed necessary by the Editors, your manuscript will be sent back to one or more of the original reviewers for assessment. If the original reviewers are not available, we may invite new reviewers.

- Data accessibility

<http://datadryad.org/submit?journalID=RSOS&manu=RSOS-200233>

- Competing interests

- Authors' contributions

- Acknowledgements

- Funding statement

on behalf of Dr Isabelle Mareschal (Associate Editor)
openscience@royalsociety.org

Associate Editor's comments (Dr Isabelle Mareschal):

Two expert reviewers have read your paper and provided very helpful suggestions to improve it, largely centred around the analysis though one reviewer also points out the problem of sample size which would need to be addressed.

Comments to Author:

Reviewers' Comments to Author:

Reviewer: 1

Comments to the Author(s)

This manuscript presents an analysis of a face processing battery. The authors examine the performance of over 140 individuals on multiple tests of face matching and face memory, to determine the level of consistency between and within different sub-processes of face recognition. Their analysis focuses on the group level associations, as well as associations at the extreme ends

of the ability spectrum. While many of the tests show modest correlations, the findings indicate substantial heterogeneity in performance, even for high/low performers who are tested on tasks which (in theory) tap into similar processes.

Overall, I found the paper well-written. The tasks and analysis were described clearly, and the introduction and discussion addressed a number of relevant issues around the use of screening batteries in applied settings. I feel that the manuscript could be strengthened by considering the psychometric properties and different trial types within the tests in more detail, and considering some of the theoretical implications of their finding in addition to the more practical/policy-based implications.

Major comments:

1) Overall, I think the approach that the authors have taken in the results section is interesting and appropriate to their question. However, I believe that the reliability of the different tests (or even sub-components of the tests, see below) is an extremely important consideration, and the authors should (where possible) calculate or report this information. This is particularly important when the tests are being used to assess individual differences, and where correlations are being calculated. If tests have relatively low reliability, that can constrain the maximum correlation between tests – for example, if two tests have reliabilities of .86 and .40, the maximum correlation is around .59. If this is the case, a correlation of .35 is quite impressive! You can also use reliability to get attenuated correlations – this may provide a clearer estimate of the real relationship between the different measures. Bowles et al. (2009; doi:10.1080/02643290903343149) include a good example of these concepts applied to face recognition tasks.

While reliability estimates may be difficult for some of the tests in the battery (the sorting task is an interesting challenge), they should be fairly straightforward for most of the others. Including an estimate of reliability is also very important if the authors want to claim that their new, shortened Yearbook Test has appropriate psychometric properties for assessing individual differences.

2) I'm a bit confused as to why the IE measure was analysed for one measure but not any others. What is the justification for this? Presumably it isn't purely mathematical, since the IE doesn't meet the requirements outlined by Bruyer & Brysbaert (2011). While RT-based measures can provide an interesting additional perspective on face processing, and might be relevant for understanding different aspects of face cognition, I would suggest that the authors rethink their use of IE for the KFMT alone, and either examine correct RTs for all the tasks where that is appropriate/accessible, or remove RT-based measures from consideration entirely.

3) I am also concerned that performance in target-present/hit trials and target-absent/correct-rejection trials is being conflated in several tasks, particularly the 1-in-10 and MMT. There's a reasonable amount of evidence that performance on these trials tends to dissociate (e.g., Bate et al. 2019, DOI: 10.1002/acp.3525 for a recent example), and given the focus of the paper (i.e., examining consistency across different sub-processes of face processing) this seems an unusual omission. Even if/when the overall performance is of interest (and most of the time I believe the tasks should be broken down to separate the TA/TP trials – this has the potential to be far more informative at both the group and individual level), then the appropriate measure in these tasks is d' or A , which account for bias in responding.

4) The introduction and discussion would benefit from a more theoretical orientation. For example, the point that some tasks are more likely to index viewpoint-invariant processing whereas others can be solved using a more feature-based strategy is interesting and relevant to face processing models (and might be explored further using RT/IE-based measures).

Minor comments:

1) I found the use of "face recognition" as distinct from "face matching" quite confusing terminology (as typically when you're matching a face on the basis of identity there's an element of recognition involved as well). In the abstract, the authors refer to face memory vs face matching – this is much clearer, and I think it would be a good adjustment to apply throughout.

- 2) You mention that responses were always self-paced: were participants aware that RT was being monitored in the KFMT task? If not, this provides further justification for removing the IE measure from analysis.
- 3) Figure 1A (showing the YBT-10) doesn't seem to align with the textual description of the task – it seems that the figure shows the to-be-matched faces on top, rather than on the left?
- 4) I'm pretty sure the MMT doesn't ask people to remember 102 faces...? It's difficult, but not impossible!
- 5) Given the sheer number of correlations being conducted (and the conservative nature of a straight Bonferroni correction), have the authors considered Holmes' sequential Bonferroni procedure when correcting for multiple comparisons?
- 6) What were the correlations between IE for the KFMT and other measures? These should be reported in-text or included in the correlation table
- 7) I found Table 2 very hard to read and interpret. Perhaps the authors could reconfigure it to z-scores instead of raw scores? This would enable much easier comparisons between mean performance on different tests.
- 8) The authors mention that there were two poor performers whose performance was in the bottom 5% on multiple tests – similar information would be useful for the top performers as well (something along these lines is mentioned in the discussion but it should be reported in the results).
- 9) It's difficult to say that the YBT-10 is “no less reliable” (p. 20, line 37) without calculating/reporting reliability. Certainly, the data supports convergent validity, which is great, but more work is needed to say that it's equally reliable.
- 10) I am also curious as to why the KFMT-long and the YBT-10 did not correlate, given their general similarity. Have the authors investigated their proposal (re the length of the KFMT) by examining correlations for the first section of the KFMT only (assuming the sections are presented in stable order/are equivalent difficulty)?

Reviewer: 2

Comments to the Author(s)

This study reports individual differences data from six tests of face abilities. The goal of the study is threefold: provide normative data for one test, examine correlations between tests, and compare between-test performance of the top 5% and bottom 5% performers. The authors conclude that there is little relationship between performance across tests, and so they recommend the use of multiple tests for personnel selection.

The paper is well-written and the research question is important. I agree with the authors that we need to develop better tests of face processing abilities. Such tests would help us understand the mechanisms that contribute to the abilities and select forensic/security personnel with superior face recognition skills. I think this paper can make important contributions to the literature, but I have several major concerns:

1. The sample size of this study is too small for the authors to draw solid inferences. Modelling studies suggest a sample size of 250+ is needed to estimate a stable correlation between two typical measures in psychology (e.g., Schonbrodt & Perrugini, 2013). That number is likely to increase for this study since it compares multiple correlations across six measures. The authors should increase the current sample size by a considerable extent. It may even be useful to do a power analysis based on predictions about specific patterns of results, for example a strong association between two matching tests and a weak(er) association between a matching test and a recognition test.
2. Analysis 2 is incomplete because it doesn't test whether the correlations are statistically different from each other. For example, the claim that CFMT+ and MMT+ are different because CFMT+ correlates with YBT-10 but MMT does not is based on the mistaken assumption that the difference between a significant correlation (e.g., $r = 0.28$ for CFMT+ and YBT-10) and a non-significant correlation (e.g., $r = 0.23$ for MMT and YBT-10) must be significant. To support this

claim the authors have to test whether $r=0.28$ and $r=0.23$ are statistically different, typically done using the Fisher's r to z transform.

3. How reliable are the six tests? It's difficult to make sense of individual-level analysis without knowing how much measurement noise the data contains. Maybe the tests are highly reliable, and so the weak generalisation across tests means that the tests are truly capturing different aspects of face processing abilities. But maybe the tests are noisy enough to allow meaningful comparisons of the 5% top/bottom performers. At a minimum, the authors could compute some sort of reliability indices (e.g., Cronbach's alpha) and incorporate them into their analyses and inferences.

4. Very few studies report the test-retest reliability of standardised face recognition tests (e.g., CFMT in Wilmer et al 2010). Test-retest reliability is necessary if researchers are seriously interested in developing valid and consistent measures of face abilities, especially if we want to use them in high-stake forensic/security settings. I highly encourage the authors to rerun the six tests with the same participants (or a subset of them) and compute their test-retest reliabilities. This would be a huge contribution to the field. A good model is the paper by Goodbourn et al (2012), which tests 1000+ participants with four measures of magnocellular processing and retests 100+ of them one week later.

Author's Response to Decision Letter for (RSOS-200233.R0)

See Appendix A.

RSOS-200233.R1 (Revision)

Review form: Reviewer 1

Is the manuscript scientifically sound in its present form?

Yes

Are the interpretations and conclusions justified by the results?

Yes

Is the language acceptable?

Yes

Do you have any ethical concerns with this paper?

No

Have you any concerns about statistical analyses in this paper?

No

Recommendation?

Accept with minor revision (please list in comments)

Comments to the Author(s)

Thank you to the authors for addressing the comments raised in my previous review. I have no additional major points.

Some very minor (textual) points:

- 1) In the analysis of top/bottom 5% of performers, you switch from using d' to using accuracy for the KFMT, 1-in-10, and MMT. This is only apparent after a paragraph or so of text, or by examining the tables in detail. Could the authors consider stating this switch in measures more clearly at the beginning of the section, to allay confusion in readers who aren't familiar with the tests?
- 2) Was IE still used in the ranking procedure (as stated on page 15) or was this overlooked in the amendments to the manuscript?
- 3) In the discussion you state that "Second, it is sensible to determine the reliability of a test designed to screen for superior skill among a group of normal observers" and go on to suggest potentially higher reliability in superior recognisers. Do you mean it is *not* sensible to determine reliability in typical observers?

Review form: Reviewer 2

Is the manuscript scientifically sound in its present form?

Yes

Are the interpretations and conclusions justified by the results?

Yes

Is the language acceptable?

Yes

Do you have any ethical concerns with this paper?

Yes

Have you any concerns about statistical analyses in this paper?

No

Recommendation?

Accept with minor revision (please list in comments)

Comments to the Author(s)

I thank the authors for the revision. I think the paper is now stronger and mostly ready for publication. I have two lingering points that should be straightforward to address:

1. I'm well aware that this study tested a larger N than similar published studies, but it seems to me that $N=146$ may still not be enough to estimate robust/stable correlations across multiple tasks. I understand the reason for not collecting more data, so my suggestion would be to add 1-2 sentences about this issue in the discussion. The authors could mention the strength of their study (i.e., testing a larger sample than most prior studies), but also the potential limitation that the reported correlations may be a bit noisy (i.e., with reference to the paper that recommends $N>250$). This information would help readers calibrate their evaluation of the evidence in this study.

2. I previously said "incomplete analysis". This wasn't a good choice of term, so I apologise for any misunderstandings. My point was simply that to make any inferences based on correlations values, one would have to test whether the correlation values are statistically different. For example, page 25 of the revised paper says "The KFMT-long, for instance, correlated moderately with the CMFT+, but associated weakly with the MMT." The distinction between "moderate" and "weak" implies that the KFMT-long/CFMT correlation is statistically stronger than the KFMT-long/MMT correlation. But my calculations suggest that the two

correlations are statistically indistinguishable, z -score = 1.11, $p > .1$, so I don't think the moderate/weak distinction here is warranted. My reading of the correlation matrix suggests that there is a positive manifold across all tasks, with little if any evidence of dissociations between subprocesses of face cognition. To me this would be a novel contribution of this paper – the notion that face matching and face memory may share more common processes than previously thought.

Decision letter (RSOS-200233.R1)

Dear Dr Ramon:

On behalf of the Editors, I am pleased to inform you that your Manuscript RSOS-200233.R1 entitled "Differences between and within individuals, and sub-processes of face cognition – implications for theory, research, and personnel selection" has been accepted for publication in Royal Society Open Science subject to minor revision in accordance with the referee suggestions. Please find the referees' comments at the end of this email.

The reviewers and Subject Editor have recommended publication, but also suggest some minor revisions to your manuscript. Therefore, I invite you to respond to the comments and revise your manuscript.

- Ethics statement

- Data accessibility

If you wish to submit your supporting data or code to Dryad (<http://datadryad.org/>), or modify your current submission to dryad, please use the following link:
<http://datadryad.org/submit?journalID=RSOS&manu=RSOS-200233.R1>

- Competing interests

- Authors' contributions

All submissions, other than those with a single author, must include an Authors' Contributions section which individually lists the specific contribution of each author. The list of Authors

should meet all of the following criteria; 1) substantial contributions to conception and design, or acquisition of data, or analysis and interpretation of data; 2) drafting the article or revising it critically for important intellectual content; and 3) final approval of the version to be published.

- Acknowledgements

- Funding statement

Because the schedule for publication is very tight, it is a condition of publication that you submit the revised version of your manuscript before 06-Aug-2020. Please note that the revision deadline will expire at 00.00am on this date. If you do not think you will be able to meet this date please let me know immediately.

- 1) A text file of the manuscript (tex, txt, rtf, docx or doc), references, tables (including captions) and figure captions. Do not upload a PDF as your "Main Document".
- 2) A separate electronic file of each figure (EPS or print-quality PDF preferred (either format should be produced directly from original creation package), or original software format)
- 3) Included a 100 word media summary of your paper when requested at submission. Please ensure you have entered correct contact details (email, institution and telephone) in your user account
- 4) Included the raw data to support the claims made in your paper. You can either include your data as electronic supplementary material or upload to a repository and include the relevant doi within your manuscript

5) All supplementary materials accompanying an accepted article will be treated as in their final form. Note that the Royal Society will neither edit nor typeset supplementary material and it will be hosted as provided. Please ensure that the supplementary material includes the paper details where possible (authors, article title, journal name).

on behalf of Dr Isabelle Mareschal (Associate Editor)
openscience@royalsociety.org

Associate Editor Comments to Author (Dr Isabelle Mareschal):

Associate Editor: 1

Comments to the Author:

Both reviewers are satisfied with the revisions but raise a few (minor) points that should be straightforward to address.

Reviewer comments to Author:

Reviewer: 1

Comments to the Author(s)

Thank you to the authors for addressing the comments raised in my previous review. I have no additional major points.

Some very minor (textual) points:

- 1) In the analysis of top/bottom 5% of performers, you switch from using d' to using accuracy for the KFMT, 1-in-10, and MMT. This is only apparent after a paragraph or so of text, or by examining the tables in detail. Could the authors consider stating this switch in measures more clearly at the beginning of the section, to allay confusion in readers who aren't familiar with the tests?
- 2) Was IE still used in the ranking procedure (as stated on page 15) or was this overlooked in the amendments to the manuscript?
- 3) In the discussion you state that "Second, it is sensible to determine the reliability of a test designed to screen for superior skill among a group of normal observers" and go on to suggest potentially higher reliability in superior recognisers. Do you mean it is *not* sensible to determine reliability in typical observers?

Reviewer: 2

Comments to the Author(s)

I thank the authors for the revision. I think the paper is now stronger and mostly ready for publication. I have two lingering points that should be straightforward to address:

1. I'm well aware that this study tested a larger N than similar published studies, but it seems to me that N=146 may still not be enough to estimate robust/stable correlations across multiple tasks. I understand the reason for not collecting more data, so my suggestion would be to add 1-2 sentences about this issue in the discussion. The authors could mention the strength of their study (i.e., testing a larger sample than most prior studies), but also the potential limitation that the reported correlations may be a bit noisy (i.e., with reference to the paper that recommends $N > 250$). This information would help readers calibrate their evaluation of the evidence in this study.

2. I previously said "incomplete analysis". This wasn't a good choice of term, so I apologise for any misunderstandings. My point was simply that to make any inferences based on correlations values, one would have to test whether the correlation values are statistically different. For example, page 25 of the revised paper says "The KFMT-long, for instance, correlated moderately with the CMFT+, but associated weakly with the MMT." The distinction between "moderate" and "weak" implies that the KFMT-long/CFMT correlation is statistically stronger than the KFMT-long/MMT correlation. But my calculations suggest that the two correlations are statistically indistinguishable, $z\text{-score} = 1.11$, $p > .1$, so I don't think the moderate/weak distinction here is warranted. My reading of the correlation matrix suggests that there is a positive manifold across all tasks, with little if any evidence of dissociations between subprocesses of face cognition. To me this would be a novel contribution of this paper - the notion that face matching and face memory may share more common processes than previously thought.

Author's Response to Decision Letter for (RSOS-200233.R1)

See Appendix B.

Decision letter (RSOS-200233.R2)

Dear Dr Ramon,

It is a pleasure to accept your manuscript entitled "Differences between and within individuals, and sub-processes of face cognition - implications for theory, research, and personnel selection" in its current form for publication in Royal Society Open Science.

Due to rapid publication and an extremely tight schedule, if comments are not received, your paper may experience a delay in publication. Royal Society Open Science operates under a continuous publication model. Your article will be published straight into the next open issue and this will be the final version of the paper. As such, it can be cited immediately by other researchers. As the issue version of your paper will be the only version to be published I would

advise you to check your proofs thoroughly as changes cannot be made once the paper is published.

on behalf of Dr Isabelle Mareschal (Associate Editor)

Appendix A

Responses to Reviewer 1's comments

We thank Reviewer 1 for the favourable and generally positive assessment of our work, as well as the constructive comments, which we have addressed as detailed below.

Major comments

1) "Overall, I think the approach that the authors have taken in the results section is interesting and appropriate to their question. However, I believe that the reliability of the different tests (or even sub-components of the tests, see below) is an extremely important consideration, and the authors should (where possible) calculate or report this information. This is particularly important when the tests are being used to assess individual differences, and where correlations are being calculated. If tests have relatively low reliability, that can constrain the maximum correlation between tests – for example, if two tests have reliabilities of .86 and .40, the maximum correlation is around .59. If this is the case, a correlation of .35 is quite impressive! You can also use reliability to get attenuated correlations – this may provide a clearer estimate of the real relationship between the different measures. Bowles et al. (2009; doi:10.1080/02643290903343149) include a good example of these concepts applied to face recognition tasks. While reliability estimates may be difficult for some of the tests in the battery (the sorting task is an interesting challenge), they should be fairly straightforward for most of the others. Including an estimate of reliability is also very important if the authors want to claim that their new, shortened Yearbook Test has appropriate psychometric properties for assessing individual differences."

We agree with the recommendation of establishing the reliability of the tests that we have used (where possible), and have reported the reliability of each test in-text, with the exception of the FICST (where it is not possible due to its procedure). In addition to reliability measured for performance across trials, we now also report test-retest reliability for the YBT-10 based on data acquired online, from an independent cohort, who performed this test twice with a minimum 6-day inter-session interval.

2) "I'm a bit confused as to why the IE measure was analysed for one measure but not any others. What is the justification for this? Presumably it isn't purely mathematical, since the IE doesn't meet the requirements outlined by Bruyer & Brysbaert (2011). While RT-based measures can provide an interesting additional perspective on face processing, and might be relevant for understanding different aspects of face cognition, I would suggest that the authors rethink their use of IE for the KFMT alone, and either examine correct RTs for all the tasks where that is appropriate/accessible, or remove RT-based measures from consideration entirely."

Our reasoning for originally reporting IE was to consider both RTs and accuracy, since e.g. passport control requires the ability to process queues both quickly *and* accurately. The rationale for providing this measure for the KFMT alone was due to the fact that it is the only test requiring 2AFC choices, where observers are more likely to exhibit speed-accuracy trade-offs. Following the reviewer's comments, we have removed the IE measure from the main document, and have relocated it to the Supplementary Material for any readers who may wish to view this information.

3) "I am also concerned that performance in target-present/hit trials and target-absent/correct-rejection trials is being conflated in several tasks, particularly the 1-in-10 and MMT. There's a reasonable amount of evidence that performance on these trials tends to dissociate (e.g., Bate et al. 2019, DOI: 10.1002/acp.3525 for a recent example), and given the focus of the paper (i.e., examining consistency across different sub-processes of face processing) this seems an unusual omission. Even if/when the overall performance is of interest (and most of the time I believe the tasks should be broken down to separate the TA/TP trials – this has the potential to be far more informative at both the group and individual level), then the appropriate measure in these tasks is d' or A , which account for bias in responding."

The issue of conflating accuracy on target-present/match trials with that on target-absent/mismatch ones is a tricky one. On the one hand, match and mismatch accuracy do, indeed, dissociate (e.g., Megreya &

Burton, 2007). However, it is also not uncommon for papers to conflate the two when correlating performance between tests (e.g., McCaffery, Robertson, Young, & Burton, 2018; Robertson, Noyes, Dowsett, Jenkins, & Burton, 2016; Verhallen et al., 2017). Given the number of correlations (and subsequent adjustments for multiple comparisons) that the separate consideration of these trial types would entail (48 in total), we decided to follow the reviewer's recommendation to use sensitivity (i.e. d') as our overall performance measure. However, we provide trial-type dependent analyses in the Supplementary Material.

4) "The introduction and discussion would benefit from a more theoretical orientation. For example, the point that some tasks are more likely to index viewpoint-invariant processing whereas others can be solved using a more feature-based strategy is interesting and relevant to face processing models (and might be explored further using RT/IE-based measures)."

Following the reviewer's recommendation, we have expanded the introduction and discussion sections to accommodate theoretical justifications for our work.

Minor comments

1) "I found the use of "face recognition" as distinct from "face matching" quite confusing terminology (as typically when you're matching a face on the basis of identity there's an element of recognition involved as well). In the abstract, the authors refer to face memory vs face matching – this is much clearer, and I think it would be a good adjustment to apply throughout."

Thank you very much for pointing out this important point. We have checked and adjusted the terminology throughout the manuscript, using *perception* and *recognition* to describe distinct sub-processes of face cognition. The terms *matching* and *memory*, on the other hand, describe the nature of tasks performed by observers to probe a given sub-process.

2) "You mention that responses were always self-paced: were participants aware that RT was being monitored in the KFMT task? If not, this provides further justification for removing the IE measure from analysis."

Participants in the KFMT were aware that response accuracy and RTs were being monitored (see also Major Comment 2).

3) "Figure 1A (showing the YBT-10) doesn't seem to align with the textual description of the task – it seems that the figure shows the to-be-matched faces on top, rather than on the left?"

Thank you for pointing this out; we have modified the caption to make clear that Figure 1A is an illustration of the YBT task, rather than a direct copy.

4) "I'm pretty sure the MMT doesn't ask people to remember 102 faces...? It's difficult, but not impossible!"

We have modified the method section to reflect the number of trials and task demands of the MMT.

5) "Given the sheer number of correlations being conducted (and the conservative nature of a straight Bonferroni correction), have the authors considered Holmes' sequential Bonferroni procedure when correcting for multiple comparisons?"

We have utilised Holms' Sequential Bonferroni adjustment over the straightforward Bonferroni adjustment that was previously employed.

6) "What were the correlations between IE for the KFMT and other measures? These should be reported in-text or included in the correlation table"

As mentioned above, we have removed the IE measure from the main document and provide this information in the Supplementary Material.

7) “I found Table 2 very hard to read and interpret. Perhaps the authors could reconfigure it to z-scores instead of raw scores? This would enable much easier comparisons between mean performance on different tests.”

We have amended Table 2 such that it now provides the values that reported originally, as well as z-scores such that each test can be compared along the same scale.

8) “The authors mention that there were two poor performers whose performance was in the bottom 5% on multiple tests – similar information would be useful for the top performers as well (something along these lines is mentioned in the discussion but it should be reported in the results).”

Indeed, in the manuscript we indicate(d) that *“In addition, no observers scored among the top 5% across all six tests; only one observer fell into this bracket across three tests (YBT-10, FICST, and CFMT+).”*

9) “It’s difficult to say that the YBT-10 is “no less reliable” (p. 20, line 37) without calculating/reporting reliability. Certainly, the data supports convergent validity, which is great, but more work is needed to say that it’s equally reliable.”

We agree with comment and have modified our phrasing so as to not imply that the YBT-10 is more or less reliable than the its longer counterpart, as reliability for the long version is not reported in Stacchi et al. (2020). However, as mentioned in response to Major Comment 1, we now provide indices of reliability for the YBT-10 (as well as other tests).

10) “I am also curious as to why the KFMT-long and the YBT-10 did not correlate, given their general similarity. Have the authors investigated their proposal (re the length of the KFMT) by examining correlations for the first section of the KFMT only (assuming the sections are presented in stable order/are equivalent difficulty)?”

Following our usage of Holm’s Sequential Bonferroni adjustment, the KFMT and the YBT-10 now correlate.

Responses to Reviewer 2's comments

We thank Reviewer 2 for the positive assessment of our manuscript, acknowledging the importance of using more ecologically-valid measures of face processing, and providing constructive suggestions to improve our work as detailed below.

1. "The sample size of this study is too small for the authors to draw solid inferences. Modelling studies suggest a sample size of 250+ is needed to estimate a stable correlation between two typical measures in psychology (e.g., Schonbrodt & Ferrugini, 2013). That number is likely to increase for this study since it compares multiple correlations across six measures. The authors should increase the current sample size by a considerable extent. It may even be useful to do a power analysis based on predictions about specific patterns of results, for example a strong association between two matching tests and a weak(er) association between a matching test and a recognition test."

Thank you for pointing out this important paper. We have decided against increasing our sample size to increase the power of our correlations for two primary reasons.

First and foremost, we are currently unable to gain access to laboratories, experimenters, and volunteers for human subject testing. We are reluctant to collect additional data online given that this would mean relinquishing control over the testing environment, which may produce unwanted variation within our sample.

Second, our sample is already quite large, and exceeds that of many recent correlational studies (see, Balsdon, Summersby, Kemp, & White, 2018 ($n = 114$); Fysh & Bindemann, 2018 ($n = 50$); Fysh, 2018 ($n = 70$); McCaffery et al., 2018 ($n = 103$); Robertson et al., 2017 ($n = 40$)). The results of these studies converge with our findings, that is to say they detect correlations between different face matching tests (Balsdon et al., 2018; Fysh & Bindemann, 2018), as well as between face matching and face memory (Fysh & Bindemann, 2018; McCaffery et al., 2018; Robertson et al., 2017). Given that these correlations converge across studies, we are confident that our results are reflective of true relationships between tasks.

2. "Analysis 2 is incomplete because it doesn't test whether the correlations are statistically different from each other. For example, the claim that CFMT+ and MMT+ are different because CFMT+ correlates with YBT-10 but MMT does not is based on the mistaken assumption that the difference between a significant correlation (e.g., $r = 0.28$ for CFMT+ and YBT-10) and a non-significant correlation (e.g., $r = 0.23$ for MMT and YBT-10) must be significant. To support this claim the authors have to test whether $r=0.28$ and $r=0.23$ are statistically different, typically done using the Fisher's r to z transform."

We disagree with the suggestion that Analysis 2 is "incomplete". The aim of the paper was never to establish whether one pair of tests correlated more strongly than another pair of tests, although we appreciate how this impression may have been miscommunicated. To address this, we have amended our wording in the General Discussion so as to not imply that some correlations are stronger or weaker than others.

3. "How reliable are the six tests? It's difficult to make sense of individual-level analysis without knowing how much measurement noise the data contains. Maybe the tests are highly reliable, and so the weak generalisation across tests means that the tests are truly capturing different aspects of face processing abilities. But maybe the tests are noisy enough to allow meaningful comparisons of the 5% top/bottom performers. At a minimum, the authors could compute some sort of reliability indices (e.g., Cronbach's alpha) and incorporate them into their analyses and inferences."

This is a valid point, which we have addressed in the revised manuscript. We now report reliability of the computer-based tests in text, and additional reliability analyses for the YBT-10 (see next response below).

4. “Very few studies report the test-retest reliability of standardised face recognition tests (e.g., CFMT in Wilmer et al 2010). Test-retest reliability is necessary if researchers are seriously interested in developing valid and consistent measures of face abilities, especially if we want to use them in high-stake forensic/security settings. I highly encourage the authors to rerun the six tests with the same participants (or a subset of them) and compute their test-retest reliabilities. This would be a huge contribution to the field. A good model is the paper by Goodbourn et al (2012), which tests 1000+ participants with four measures of magnocellular processing and retests 100+ of them one week later.”

We recognise the value in acquiring test-retest reliability for the measures reported in the paper, and agree that this would be a large contribution to the field. However, in line with our response to Major Point 1, we have not done this due to the fact that the current pandemic restricts us from testing our cohort a second time.

However, we agree that for our new test the YBT-10, establishing reliability is important. Therefore, we have implemented the digital analogue to the original, paper-based version. As detailed in the revised manuscript (see also raw data), we recruited an independent sample who performed this test twice online, with a minimum inter-session interval of 6 days (for a similar approach with the KFMT, see also Fysh & Bindemann, 2018). We now report reliability for both modalities, together with test-retest reliability for the YBT-10 online version. Thanks to the reviewers’ comments, this provided additional insights, which we discuss and will address in the future.

Faculté des Lettres
Applied Face Cognition Lab
Département de Psychologie
Dr. Meike Ramon
Rue P.A.-de-Faucigny 2
CH1700 Fribourg

T +41 26 300 7533
meike.ramon@unifr.ch

Bern, 31 July 2020

Dear Dr. Mareschal,

Please find attached the final version of our manuscript *Differences between and within individuals, and sub-processes of face cognition – implications for theory, research, and personnel selection*.

We are pleased that both reviewers are satisfied with our previous modifications. We thank them for their final helpful suggestions. You will find these highlighted in the revised document, with no further response to the reviewers' comments provided.

Thank you for your editorial work and accepting our work for publication in *Royal Society Open Science*. On behalf of all the authors

Sincerely,

Meike Ramon

SNSF PRIMA Fellow & Group Leader